# Spatial distribution and determinants of abortion among reproductive age women in Ethiopia, evidence from Ethiopian Demographic and Health Survey 2016 data: Spatial and mixed-effect analysis

**Getayeneh Antehunegn Tesema**[1]*, **Tesfaye Hambisa Mekonnen**[2], **Achamyeleh Birhanu Teshale**[1]

**1** Department of Epidemiology and Biostatistics, Institute of Public Health, College of Medicine and Health Sciences, University of Gondar, Gondar, Ethiopia, **2** Department of Environmental and Occupational Health and Safety, Institute of Public Health, College of Medicine and Health Sciences, University of Gondar, Gondar, Ethiopia

* getayenehantehunegn@gmail.com

**Data Availability Statement:** The data we used for this study is available in the DHS program. A letter of approval for the use of the data was secured

## Abstract

### Background

Unsafe abortion remains a global public health concern and it is the leading cause of maternal mortality and morbidity. Despite the efforts made to improve maternal health care service utilization, unsafe abortion yet constitutes the highest maternal mortality in Sub-Saharan Africa (SSA) including Ethiopia. Although abortion among reproductive-age women is a common problem in Ethiopia, there is limited evidence about the spatial distribution and determinants of abortion. Therefore, this study aimed to investigate the spatial distribution and determinants of abortion among reproductive-age women in Ethiopia.

### Methods

A secondary data analysis was conducted using the 2016 Ethiopian Demographic and Health Survey (EDHS) data. A total of 12378 reproductive-age women were included in this study. The Bernoulli model was fitted using SaTScan version 9.6 statistical software to identify significant hotspot areas of abortion and ArcGIS version 10.6 statistical software was used to explore the spatial distributions of abortion. For the determinant factors, a mixed effect logistic regression model was fitted to take into account the hierarchical nature of the EDHS data. Deviance (-2LL), AIC, BIC, and ICC were used for model comparison. The AOR with a 95% CI was estimated for the potential determinants of abortion.

### Results

The overall prevalence of abortion in Ethiopia was 8.9% ranging from 4.5% in Benishangul to 11.3% in Tigray regions. The spatial analysis revealed that abortion was significantly varied across the country. The SaTScan analysis identified a total of 60 significant clusters, of

from the Measure DHS program and the data set was downloaded from the website www.measuredhs.com (https://dhsprogram.com/data/available-datasets.cfm). We used EDHS 2016 Birth data set (BR file) and extracted the outcome variable (abortion) and explanatory variables. The location data (latitude and longitude coordinates) was also taken from selected enumeration areas (clusters).

**Funding:** For this study, no specific funding was received from any organization since the study was based on EDHS data available in DHS program.

**Competing interests:** The authors have declared that no competing interests exist.

**Abbreviations:** AIC, Akakie Information Criteria; AOR, Adjusted Odds Ratio; BIC, Bayesian Information Criterion; CI, Confidence Interval; COR, Crude Odds Ratio; DHS, Demographic and Health Survey; DM, Diabetic Mellitus; EAs, Enumeration Areas; EDHS, Ethiopian Demographic and Health Survey; GLMM, Generalized Linear Mixed Model; HTN, Hypertension; ICC, Intra cluster correlation coefficient; IUGR, Intra uterine growth restriction; LLR, Log Likelihood Ratio; LR, Likelihood Ratio; RR, Relative risk; SNNPR, Southern nations, nationalities and people's region; SSA, Sub-Saharan Africa.

these 19 clusters were primary clusters. The primary clusters were located in the northern part of the Tigray region (LLR = 26.6, p<0.01; RR = 2.63). In the multivariable mixed-effect logistic regression analysis; primary education [AOR = 1.36; 95% CI: 1.13, 1.64], rural residence [AOR = 4.96; 95% CI: 3.42, 7.18], protestant religion follower [AOR = 0.56; 95% CI: 0.42, 0.75], richest wealth status [AOR = 1.72; 95% CI: 1.24, 2.40], maternal age 45–49 years [AOR = 3.12; 95% CI: 1.52, 6.44], listening radio [AOR = 1.27; 1.01, 1.60], and watching television [AOR = 1.45; 1.04, 2.01] were significant determinants of abortion.

## Conclusions

The prevalence of abortion remains unacceptably high in Ethiopia. The spatial distribution of abortion has been significantly varied across regions in Ethiopia. Having primary education, being rural, having media exposure, and being from the richest household were significantly associated with higher odds of abortion whereas being protestant religious followers were associated with lower odds of abortion. Therefore, the government should design public health programs targeting the identified hotspot areas of abortion and should scale up maternal health programs in rural areas, to reduce maternal morbidity and mortality.

## Background

Abortion is defined as the loss of product of conception (whether induced or spontaneous) before 28 completed weeks of gestation [1, 2]. Globally, an estimated 55.9 million unsafe abortions occur annually, of these 49.3 million were occurred in developing countries [3]. Unsafe abortion is the leading cause of maternal mortality and morbidity [4]. It accounts for 13% of global maternal mortality [5] and 5 million disabilities annually [6, 7]. The majority of unsafe abortion can be prevented through education on sexual behavior, family planning, and the provision of safe abortion [8].

Unsafe abortion is a major public health concern [3], particularly in developing countries where unintended pregnancies are common due to ineffective use or nonuse of contraceptives [9]. The magnitude of unsafe abortion has varied across countries, ranging from 3.1% in western Africa to 3.8% in northern Africa [10, 11]. Even though unsafe abortion is reduced in developed nations where the liberalization of abortion law and safe abortion service is legally available [12, 13], it remains high in developing countries particularly in Sub-Saharan Africa (SSA) where abortion is legally restricted [14–16].

Prior studies have documented that unsafe abortion has been an important and ongoing health problem in Ethiopia. In 2008, an estimated 382,000 induced abortions were performed in Ethiopia with a prevalence of 13% [6], mainly related to unwanted pregnancies [17]. According to the Ethiopian Demographic and Health Survey (EDHS) 2016 report, the maternal mortality rate was 412 per 100,000 births [18].

Previous studies done on abortion revealed that residence, parity, educational status, antenatal care (ANC) utilization, place of delivery, maternal nutritional status, and maternal obstetric factors were significantly associated with abortion [19–21]. The prevalence of abortion has been varied not only among countries but also within the country [22] and it is highly concentrated among rural residents, poor and marginalized societies [23, 24]. Thus, exploring the spatial distributions of abortion has become fundamental to design evidence-based public health interventions [25].

Though there are studies conducted on the determinants of abortion in Ethiopia [26], information is scant on the spatial distribution and its determinant factors at the national level. Therefore, we aimed to investigate the spatial distribution and determinants of abortion among reproductive-age women in Ethiopia. As abortion and abortion-related mortality is an indicator of availability and quality of maternal health services [27], understanding the significant hotspot areas of abortion would help to evaluate the quality of service and access to maternal health services. Furthermore, the findings of this study could guide policymakers in designing effective public health interventions to reduce abortion and abortion-related maternal morbidity and mortality.

## Method and materials

### Study design, setting and period

Secondary data analysis was conducted based on the 2016 EDHS data. The EDHS is a nationally representative survey conducted in every five years in the nine regional states (Afar, Amhara, Benishangul-Gumuz, Gambela, Harari, Oromia, Somali, Southern Nations, Nationalities, and People's Region (SNNPR), and Tigray), and two administrative cities (Addis Ababa and Dire-Dawa) of Ethiopia [28]. In 2016, the total population of Ethiopia was 102 million, of these 43.47% were aged less than 14 years. Around 35 million Ethiopian people are living in poverty/had low socioeconomic status. The crude birth rate in Ethiopia is 36.5 per 1000 populations with a total fertility rate of 4.46. Ethiopia has a three-tire health system; primary health care unit (Primary hospital, health center, health post, primary clinic, and medium clinic), secondary health care (General hospital, specialty clinics, and specialty centers), and tertiary health care (Specialized hospital). The number of hospitals, in general, health facilities, varies from region to region [29].

### Source and sample population

The source population was all pregnant women within five years before the survey in Ethiopia, while all pregnant women in the selected enumeration areas within five years before the survey were the study population. In EDHS, a stratified two-stage cluster sampling technique was employed using the 2007 Population and Housing Census as a sampling frame. In the first stage, 645 enumeration areas (EAs) were selected with probability proportional to the EA size and with independent selection in each sampling stratum. In the second stage, on average 28 households were systematically selected. A total weighted sample of 12378 reproductive-age women was included in this study. The detailed sampling procedure exists in the full EDHS 2016 report [30].

### Variables and data collection procedure

The dependent variable for this study was "abortion", which was derived from the EDHS question "have you ever had a terminated pregnancy". The outcome variable was dichotomized as "Yes" if a woman had experienced abortion, and "No" if a woman didn't experience abortion within the study period. The independent variables included in the study were maternal age, residence, educational status, marital status, religion, frequency of watching television, frequency of listening radio, wealth status, and birth history.

The data were accessed from the DHS program official database www.measuredhs.com, after permission was granted through an online request by explaining the objective of the study. We used the EDHS 2016 birth data (BR) set. The geographic coordinate data (longitude

and latitude coordinates) was taken at the cluster/ enumeration area level after we explain the purpose of conducting the spatial distribution of abortion.

## Data management and analysis

The data were weighted using sampling weight, primary sampling unit, and strata before any statistical analysis to restore the representativeness of the survey and to take into account the sampling design and get reliable statistical estimates.

**Spatial analysis.** ArcGIS version 10.6 and SaTScan version 9.6 statistical software were used for exploring the spatial distribution, global spatial autocorrelation, spatial interpolation, and for identifying significant hotspot areas of abortion.

*Spatial autocorrelation analysis.* The spatial autocorrelation (Global Moran's I) is the correlation coefficient for the relationship between a variable and its surrounding value, it measures the overall spatial autocorrelation of abortion [31]. Moran's I is a spatial statistics used to measure spatial autocorrelation by taking the entire data set and produce a single output. The spatial autocorrelation coefficient is statistically significant when tested against the null hypothesis that the observed value differs with its expected value which is -1/ (n-1), where n is the number of points at enumeration area level for which the autocorrelation is being computed. Moran's I value ranges from-1 to 1 [32]. A value close to 1 shows a strong positive spatial autocorrelation whereas a value close to -1 shows a strong negative spatial autocorrelation. If Moran's I close to 0, it indicates that there is no spatial autocorrelation. A statistically significant Moran's I value ($p < 0.05$) can lead to rejection of the null hypothesis (abortion is randomly distributed) and indicates the presence of spatial autocorrelation.

*Spatial interpolation.* The spatial interpolation technique was used to predict abortion on the un-sampled areas in Ethiopia based on sampled measurements. There are various deterministic and geostatistical interpolation methods. Among the interpolation techniques, ordinary Kriging and empirical Bayesian Kriging are the best interpolation methods since they optimize the weight [33]. Kriging spatial interpolation method was used in this study for predicting abortion in unobserved areas since it had a small mean square error and residual. It produces smooth maps of abortion by predicting the prevalence of abortion on the un-sampled locations (enumeration areas) and it is an optimal interpolation based on regression against observed values of the surrounding data points, and weighted according to the spatial covariance values.

*Spatial scan statistical analysis.* In the spatial scan statistical analysis, Bernoulli based model was employed to identify statistically significant spatial clusters of abortion using Kuldorff's SaTScan version 9.6 software. For this study, we used a circular scanning window that moves across the study area since the elliptical window is inactive in the SaTScan software. Women who experienced abortion were taken as cases and those who didn't experience abortion were considered as controls to fit the Bernoulli model. The numbers of cases in each location had Bernoulli distribution and the model required data for cases, controls, and geographic coordinates. The default maximum spatial cluster size of <50% of the population was used, as an upper limit, since it allowed both small and large clusters to be detected and ignored clusters that contained more than the maximum limit. Selecting the cluster size of 50% of the total population is the default option for the maximum scanning window size and it is often used to search the most likely clusters with a higher value of the likelihood value. Kuldorff's indicated that a window-sized up to 50% of the population at risk can reduce negative clusters (highly sensitive), avoid missing clusters, and more likely to contain the true significant clusters than the small scanning window.

For each potential cluster, a Likelihood Ratio (LR) test statistic and the p-value was used to determine if the number of observed abortion cases within the potential cluster was significantly higher than expected or not. The scanning window with maximum likelihood was the most likely performing cluster, and the p-value was assigned to each cluster using Monte Carlo hypothesis testing by comparing the rank of the maximum likelihood from the real data with the maximum likelihood from the random datasets. The primary and secondary clusters were identified and assigned p-values and ranked based on their likelihood ratio test, based on 999 Monte Carlo replications [34].

**Mixed effect logistic regression analysis.** Cross tabulations and summary statistics were done using STATA version 14 software. The EDHS data has hierarchical nature; hence women are nested within a cluster and we expect that women within the same cluster may be more similar to each other than women in another cluster. This violates the assumption of the traditional regression model which is the independence of observations and equal variance across clusters. Therefore, an advanced statistical model is needed to take into account the between cluster variability to get a reliable standard error and unbiased estimate. Besides, since the outcome variable was binary standard logistic regression and Generalized Linear Mixed Models (GLMM) were fitted and model comparison, as well as model fitness, was done based on the Intra-Class Correlation Coefficient (ICC), Akaike Information Criteria (AIC), Bayesian Information Criteria (BIC), and Deviance values. The mixed-effect logistic regression model was the best-fitted model since it has the lowest deviance and variables with p-value <0.20 in the bi-variable analysis were considered for the multivariable mixed-effect logistic regression model. Finally, Adjusted Odds Ratio (AOR) with 95% Confidence Interval (CI) were reported and those variables with p-value <0.05 were declared to be significant factors associated with abortion. In the bi-variable mixed-effect binary logistic regression analysis; maternal age, religion, residence, wealth status, educational status, frequency of watching television, frequency of listening radio, birth history, and marital status had a p-value< 0.2 and were considered for multivariable analysis.

However, in the multivariable analysis; educational status, residence, maternal age, frequency of watching television, frequency of listening radio, and religion were significantly associated with abortion.

## Ethics consideration

Since the study was a secondary data analysis of publically available survey data from the MEASURE DHS program, ethical approval and participant consent were not necessary for this particular study. We requested DHS Program and permission was granted to download and use the data for this study from http://www.dhsprogram.com. The Institution Review Board approved procedures for DHS public-use datasets do not in any way allow respondents, households, or sample communities to be identified. There were no names of individuals or household addresses in the data file. The geographic identifiers only go down to the regional level (where regions are typically very large geographical areas encompassing several states/provinces). Each enumeration area (Primary Sampling Unit) has a PSU number in the data file, but the PSU numbers do not have any labels to indicate their names or locations. In surveys that collect GIS coordinates in the field, the coordinates are only for the enumeration area (EA) as a whole, and not for individual households, and the measured coordinates are randomly displaced within a large geographic area so that specific enumeration areas cannot be identified.

# Result

## Socio-demographic characteristics of respondents

A total of 12378 women was included in this study. Of these, 89% were rural residents, and 44.1% were lived in the Oromia region. The majority (66.8%) of women had no formal education and about 93.7% of respondents were married. The median age of respondents was 29 (IQR± 9) years (Table 1).

## Obstetric and socioeconomic characteristics of respondents

Nearly half (44.4%) of the respondents had $\geq$ 4 births, and 23.9% of women were from the poorest household. Concerning listening radio, about 73.6% of respondents had never listened to the radio (Table 2).

## Prevalence of abortion among women in Ethiopia, 2016

The overall prevalence of abortion was 8.9% [95%CI: 8.4%-9.5%] ranging from 4.5% in Benishangul-Gumuz to 11.3% in Tigray regions (Fig 1). The prevalence of abortion among rural residents was 9.2%, whereas the prevalence of abortion among urban residents was 6.7%.

## Spatial distribution of abortion

The spatial distribution of abortion showed significant spatial variation across the country with Global Moran's I value of 0.06 (p<0.001). Each point on the map represents one census enumeration area which encompasses several abortion cases. The red color indicates areas with a high prevalence of abortion, whereas the green color indicates areas with a low prevalence of abortion. In this study, the high prevalence of abortion was found in Central and Northern Tigray, Western part of Afar, Eastern part of Benishangul-Gumuz, and Southeast of SNNPRs. The low prevalence of abortion was found in the Gambela region, Western Benishangul-Gumuz, central Oromia, Harari, and Dire Dawa (Fig 2).

## Kriging interpolation of abortion

Based on EDHS 2016 sampled data, the Kriging interpolation predict the highest prevalence of abortion in Northern Tigray, Addis Ababa, Southwest Oromia, Southwest SNNPRs, and Northern Afar regions. In contrast, the relatively low prevalence of abortion was detected in Gambella, Southern part of Amhara, Western part of Benishangul-Gumuz, and Eastern part of Afar regions (Fig 3).

## Spatial scan statistical analysis

A spatial scan statistical analysis identified a total of 60 significant primary and secondary clusters. Of these 19 clusters were primary (most likely) clusters which were located in the Northern Tigray region centered at 14.175601 N, 38.891649 E with 62.42 km radius, a Relative Risk (RR) of 2.63, and Log-Likelihood Ratio (LRR) of 26.6, at p-value<0.01. It revealed that pregnant women within the spatial window had 2.63 times higher risk of experiencing abortion as compared to pregnant women outside the spatial window (Table 3). The secondary clusters were located in border areas of Oromia and Amhara regions, southeastern Oromia, and border areas between SNNPRs and Oromia regions. The bright red color circular window (Rings) indicates statistically significant spatial windows containing a high prevalence of abortion (Fig 4).

**Table 1. Socio-demographic characteristics of respondents in Ethiopia, 2016 (N = 12378).**

| Variables | Percent (%) |
|---|---|
| **Residence** | |
| Urban | 11.0 |
| Rural | 89.0 |
| **Region** | |
| Tigray | 6.4 |
| Afar | 1.1 |
| Amhara | 18.6 |
| Oromia | 44.1 |
| Somali | 4.7 |
| Ben-Gumuz | 1.1 |
| Gambela | 21.0 |
| Harari | 0.2 |
| Addis Ababa | 2.1 |
| Dire Dawa | 0.4 |
| **Maternal age (in years)** | |
| 15–19 | 3.0 |
| 20–24 | 18.0 |
| 25–29 | 30.2 |
| 30–34 | 23.2 |
| 35–39 | 16.2 |
| 40–44 | 7.0 |
| 45–49 | 2.4 |
| **Maternal educational status** | |
| No education | 66.8 |
| Primary | 26.3 |
| Secondary | 4.5 |
| Higher | 2.4 |
| **Religion** | |
| Orthodox | 34.0 |
| Muslim | 41.2 |
| Catholic | 0.9 |
| Protestant | 21.5 |
| Others* | 2.4 |
| **Husband education** | |
| No education | 45.9 |
| Primary | 36.9 |
| Secondary | 7.0 |
| Higher | 10.2 |
| **Marital status** | |
| Never married | 0.5 |
| Married | 93.7 |
| Living with a partner | 1.1 |
| Widowed | 1.2 |
| Divorced | 2.5 |
| Separated | 1.0 |

Keys:

* = Traditional religious follower.

**Table 2. Obstetric and socio-economic characteristics of participants in Ethiopia (N = 12378), 2016.**

| Variables (N = 12378) | Percentage (%) |
|---|---|
| **Wealth status** | |
| Poorest | 23.9 |
| Poor | 22.6 |
| Middle | 20.7 |
| Richer | 18.4 |
| Richest | 14.3 |
| **Frequency of listening to the radio** | |
| Not at all | 73.6 |
| Less than once a week | 13.2 |
| At least once a week | 13.3 |
| **Frequency of watching the television** | |
| Not at all | 82.1 |
| least than once a week | 10.0 |
| At least once a week | 7.9 |
| **Occupational status** | |
| Unemployed | 70.6 |
| Employed | 29.4 |
| **Birth history** | |
| No birth | 12.0 |
| One birth | 15.3 |
| Two births | 15.1 |
| Three births | 13.2 |
| Four and above births | 44.4 |
| **Preceding birth interval** | |
| Less than 24 months | 23.4 |
| $\geq$ 24 months | 76.6 |
| **Terminated pregnancy (abortion)** | |
| No | 91.1 |
| Yes | 8.9 |
| **Smoking status** | |
| Yes | 99.2 |
| No | 0.8 |

## Determinants of abortion among reproductive-age women in Ethiopia

**Model comparison.** AIC, BIC, and deviance were checked and reported as a model comparison parameter. Since the models were nested models we preferred deviance value for model comparison and the mixed effect logistic regression model was the best-fitted model because of the smallest value of deviance (Table 4). Furthermore, the ICC value which was 0.21 and the Log-likelihood ratio test which was ($X^2$ = 238.49, p-value <0.001) informed us to choose a mixed-effect logistic regression model (GLMM) over the basic model.

In the multivariable mixed-effect logistic regression model; educational status, maternal age, frequency of watching television, residence, frequency of listening radio, and religion were significantly associated with abortion.

The odds of experiencing abortion among women residing in the rural area were nearly 5 times [AOR = 4.96, 95% CI: 3.42, 7.18] higher than those residing in urban areas. The odds of experiencing abortion among women who were protestant religious followers were decreased

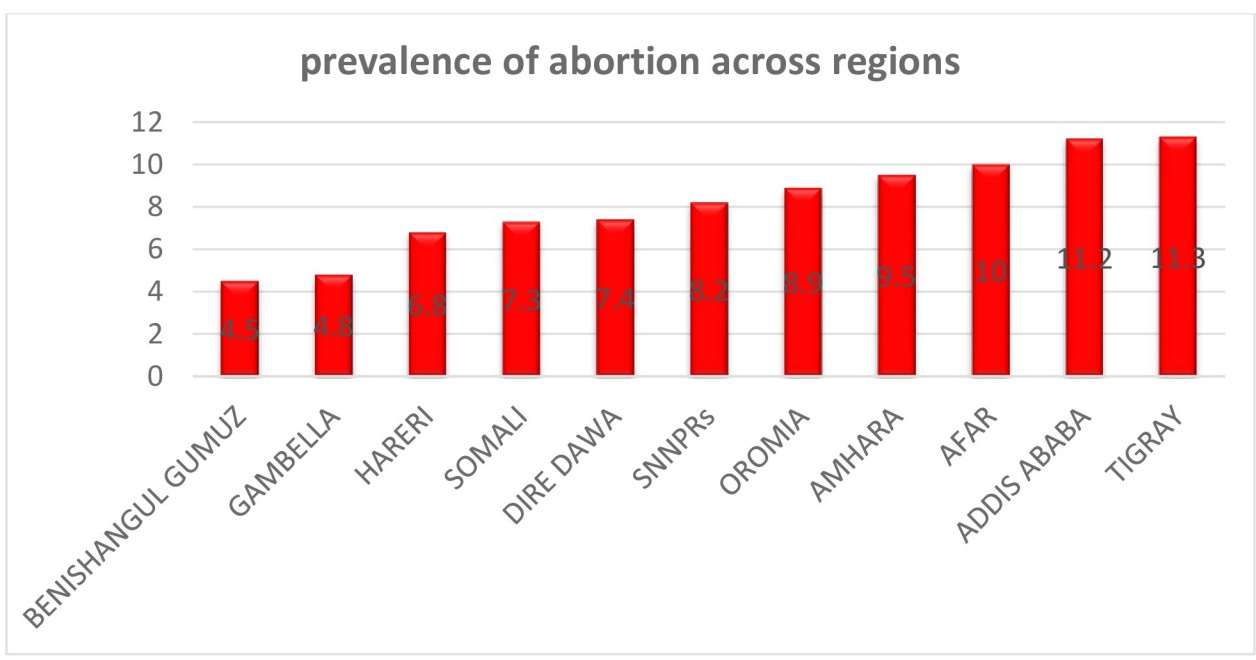

**Fig 1. Regional prevalence of abortion among reproductive-age women in Ethiopia, 2016.**

by 44% [AOR = 0.56, 95% CI: 0.42, 0.75] as compared to Orthodox Christians. The odds of experiencing abortion among women aged 24–29, 30–34, 35–39, 40–44 and 45–49 years were 2.2 times [AOR = 2.20, 95% CI: 1.27, 3.80], 3.2 times [AOR = 3.2, 95% CI: 1.82, 5.71], 3.01 times [AOR = 3.01, 95% CI: 1.67, 5.42], 4.57 times [AOR = 4.57, 95% CI: 2.47, 8.46], and 3.12 times [AOR = 3.12, 95% CI: 1.52, 6.44] higher than those women aged 15–19 years respectively. Women who attained primary education had 1.36 times [AOR = 1.36, 95% CI: 1.13, 1.64] higher odds of experiencing abortion than women who had no formal education. Women from the richest household had 1.72 times [AOR = 1.72, 95% CI: 1.24, 2.40] higher odds of experiencing abortion than women from the poorest household. Media exposure was significantly associated with abortion. The odds of having abortions among women who listened to the radio less than once a week were 1.27 times (AOR = 1.27, CI: 1.01, 1.60) higher than women who never listened to the radio. Women who watched television at least once a week had 1.45 times [AOR = 1.45, 95% CI: 1.04, 2.01] higher odds of abortion as compared to women who never watched the television (Table 5).

## Discussion

Abortion is a major public health problem in Ethiopia [35]. This study was aimed to investigate the spatial distribution and determinants of abortion in Ethiopia. The spatial analysis result revealed that the spatial distribution of abortion was significantly varied across the country. In multivariable mixed-effect logistic regression analysis; wealth status, residence, maternal education, religion, media exposure, and maternal age were significant predictors of abortion.

The current prevalence of abortion was consistent with a study reported in Mozambique [36] and lower than studies conducted in Ghana [36] and northwest Ethiopia [37]. The possible explanation might be due to the difference in the study period, study population used for the study and improvement of maternal health care service accessibility and utilization over time. But the finding of our study was found to be higher than those of studies done in India

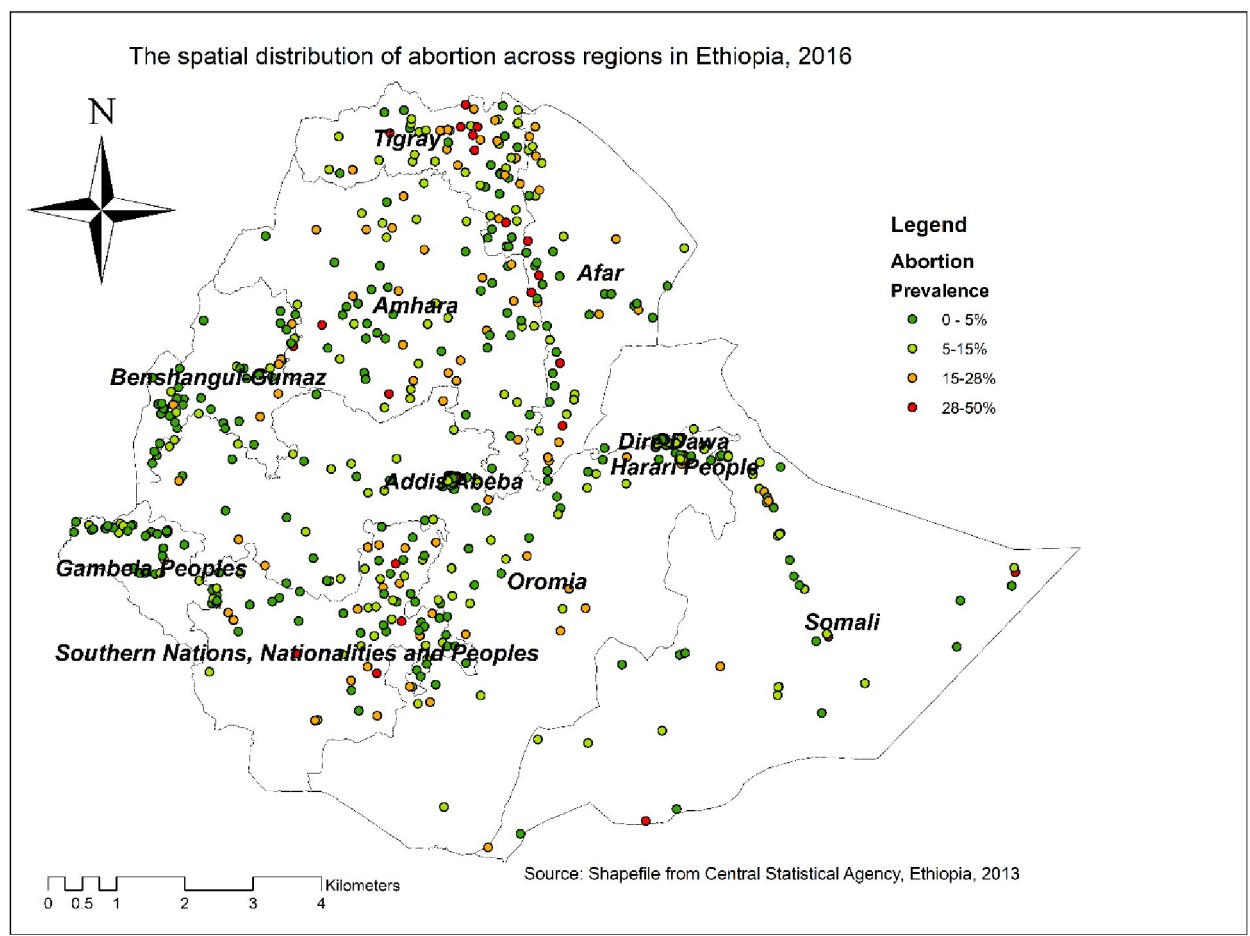

**Fig 2. The spatial distribution of abortion in Ethiopia, 2016 (source: CSA, 2013).**

(1.7%) [38] and Wolaiytasodo- Ethiopia [39]. The difference might be due to the difference in the study population. That is the current study was conducted at the national level (community-based) based on EDHS 2016 while the study in Wolaiytasodo Ethiopia was conducted among university students (institution-based) with a small sample size.

The spatial analysis result revealed that the spatial distribution of abortion was significantly varied across the country, where significant hotspot areas of abortion were identified in the northern Tigray region, border areas of Oromia, Amhara, and SNNP regions. The spatial variation might be related to the difference in socioeconomic status, and health inequality within the country. Besides, this could be attributed to the disparity in the distribution of maternal health service, and the inaccessibility of infrastructure in the border areas and the gap in health service utilization like family planning, ANC and other reproductive health services across regions [40].

In the mixed-effect logistic regression analysis, place of residence was significantly associated with abortion. Women residing in rural areas were more likely to experience abortion as compared to urban residents. It was consistent with study findings in northwest Ethiopia [37] and India [38]. This could be due to lack of access to maternal health care services utilization (such as family planning, ANC visit, awareness about danger signs of pregnancy, and birth

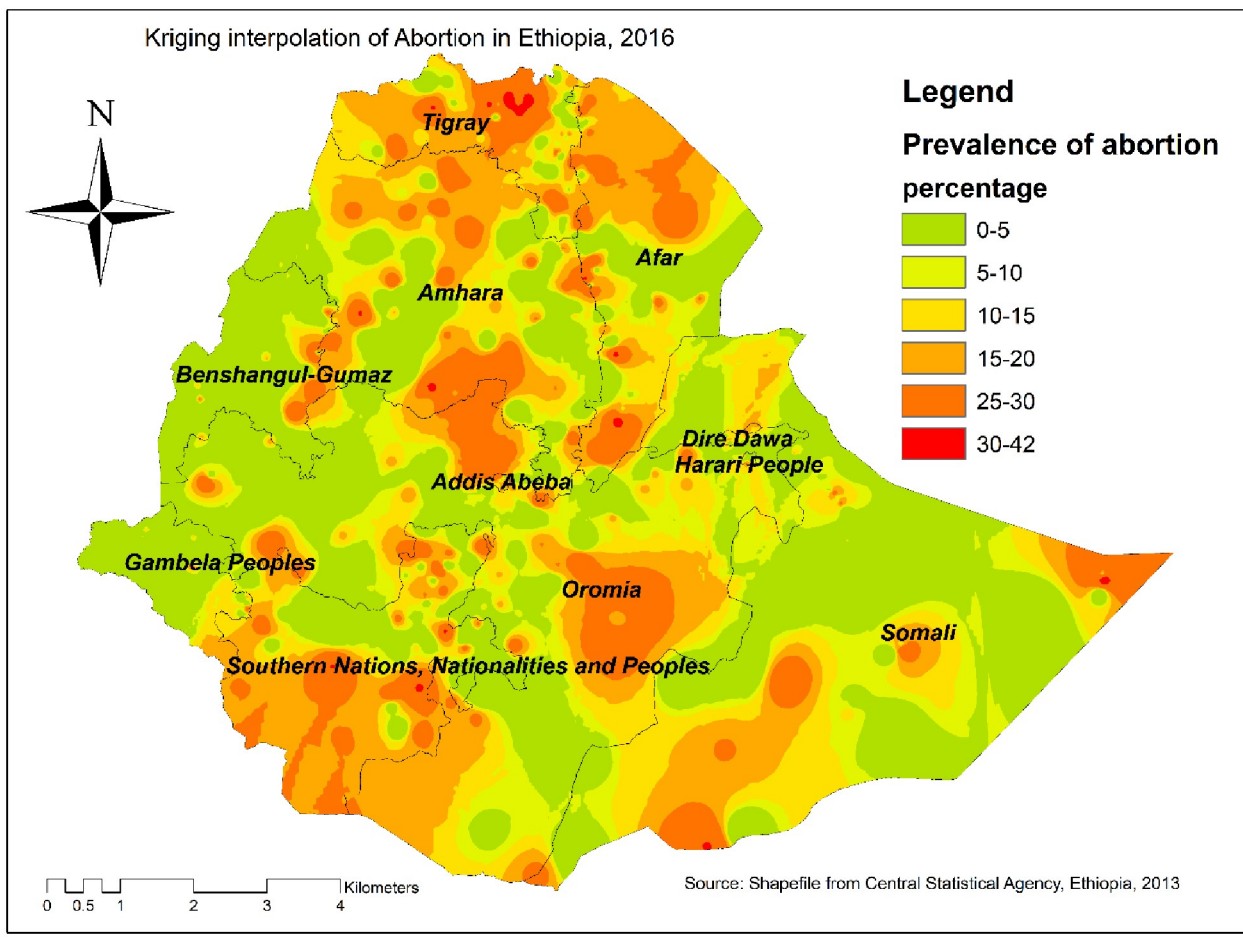

**Fig 3. The Kriging interpolation of abortion in Ethiopia, 2016 (source: CSA, 2013).**

**Table 3. Significant spatial clusters of abortion among women in Ethiopia, 2016.**

| Clusters | Enumeration areas (EAs)/ clusters detected | Coordinates/radius | Population | Cases | RR | LLR | P-value |
|---|---|---|---|---|---|---|---|
| 1 | 84, 45, 81, 590, 481, 461, 400, 636, 597, 89, 479, 604, 156, 355, 598, 584, 404, 226, 579 | (14.175601 N, 38.891649 E) / 62.42 km | 327 | 70 | 2.63 | 26.6 | <0.001 |
| 2 | 452, 472, 286, 289, 123 | (7.410925 N, 40.475707 E) / 85.79 km | 125 | 27 | 2.58 | 10.2 | 0.01 |
| 3 | 92 | (6.708449 N, 44.273542 E) / 0 km | 34 | 12 | 4.19 | 9.5 | 0.03 |
| 4 | 510, 267, 572, 10, 423, 350, 229, 482, 460, 206, 176, 531, 218, 310, 617, 120, 637, 517, 112, 201, 274, 463, 144, 464, 532, 91, 369, 170, 11, 153, 287, 339, 626, 107, 247 | (10.160658 N, 38.634847 E) / 125.60 km | 412 | 61 | 1.79 | 9.2 | 0.04 |
| 5 | 50, 342, 86, 21, 503, 450, 574, 182, 505, 398 | (5.546952 N, 37.666334 E) / 88.77 km | 267 | 42 | 1.89 | 7.5 | 0.171 |
| 6 | 276 | (10.717422 N, 40.344525 E) / 0 km | 25 | 9 | 4.26 | 7.3 | 0.218 |
| 7 | 564, 39, 230, 51 | (9.555410 N, 40.326165 E) / 34.04 km | 61 | 15 | 2.92 | 7.08 | 0.245 |

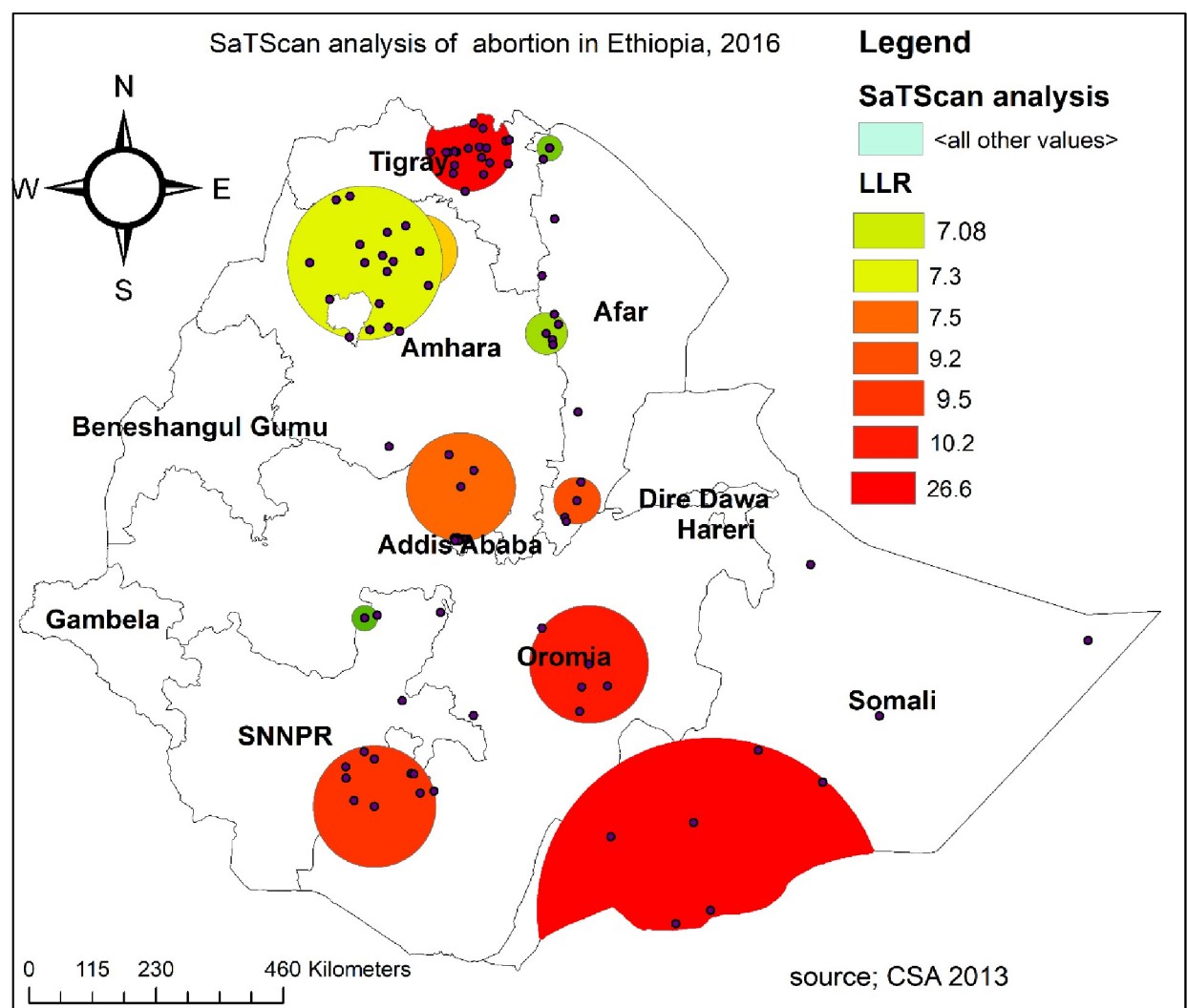

**Fig 4. The SaTScan analysis of hotspot areas of abortion in Ethiopia, 2016 (source: CSA, 2013).**

preparedness), and limited information about complications of abortion due to lack of access to media in the rural areas [41].

Maternal age was found to be significantly associated with abortion. Women in the age group 25–29, 30–34, 35–39, 40–44, and 45–49 years were more likely to experience abortion than women in the age group of 15–19 years. This was consistent with the study findings reported in Ghana [36], Denmark [42], and Mozambique [36]. The possible explanation could be because older women are more likely to have medical and pregnancy-related complications like high blood pressure (HTN), Diabetic Mellitus (DM), cervical incompetence, cardiovascular diseases and chromosomal abnormality that could complicate the pregnancy and increase

**Table 4. Model comparison between standard logistic regression and mixed-effects logistic regression.**

| Model comparison | AIC | BIC | Deviance |
|---|---|---|---|
| Logistic regression model | 6856.17 | 7077.95 | 6796.09 |
| Mixed effect logistic regression model | 6622.02 | 6851.19 | 6560.02 |

**Table 5. Multivariable mixed-effect logistic regression analysis for assessing determinants of abortion among reproductive age women in Ethiopia, 2016.**

| Variable | Abortion | | AOR (95% CI) |
|---|---|---|---|
| | No | Yes | |
| **Residence** | | | |
| Urban | 1,983 | 130 | 1 |
| Rural | 8,989 | 897 | 4.96 (3.42, 7.18) ** |
| **Age** | | | |
| 15–19 | 390 | 18 | 1 |
| 20–24 | 2,211 | 124 | 1.27 (0.74, 2.19) |
| 25–29 | 3,280 | 282 | 2.20 (1.27, 3.80) ** |
| 30–34 | 2,443 | 279 | 3.23 (1.82, 5.71) ** |
| 35–39 | 1,758 | 192 | 3.01 (1.67, 5.42) ** |
| 40–44 | 670 | 107 | 4.57 (2.47, 8.46) ** |
| 45–49 | 220 | 25 | 3.12 (1.52, 6.44) ** |
| **Wealth status** | | | |
| Poorest | 4,166 | 387 | 1 |
| Poorer | 1,848 | 149 | 0.85 (0.67, 1.07) |
| Middle | 1,490 | 154 | 1.07 (0.84, 1.36) |
| Richer | 1,361 | 128 | 0.91 (0.70, 1.19) |
| Richest | 2,107 | 209 | 1.72 (1.24, 2.40) * |
| **Educational status** | | | |
| No education | 7,158 | 670 | 1 |
| Primary | 2,688 | 269 | 1.36 (1.13, 1.64) ** |
| Secondary | 740 | 55 | 0.98 (0.68, 1.41) |
| Higher | 386 | 33 | 0.99(0.62, 1.61) |
| **Religion** | | | |
| Orthodox | 3,083 | 354 | 1 |
| Muslim | 5,647 | 518 | 0.81 (0.64, 1.01) |
| catholic | 75 | 3 | 0.40 (0.12, 1.39) |
| Protestant | 1,981 | 136 | 0.56 (0.42, 0.75) ** |
| Others | 186 | 16 | 0.66(0.34, 1.26) |
| **Frequency of listening to the radio** | | | |
| Not at all | 8,456 | 733 | 1 |
| Less than once a week | 1,265 | 147 | 1.27 (1.01, 1.60) * |
| At least once a week | 1,251 | 147 | 1.21 (0.96, 1.55) |
| **Frequency of watching television** | | | |
| Not at all | 8,754 | 791 | 1 |
| Less than once a week | 877 | 102 | 1.25 (0.95, 1.65) |
| At least once a week | 1,341 | 134 | 1.45 (1.04, 2.01) * |
| **Birth history** | | | |
| zero birth | 1,416 | 96 | 1 |
| One birth | 1,822 | 142 | 0.97 (0.72, 1.31) |
| Two births | 1,649 | 146 | 0.92(0.66, 1.27) |
| Three births | 1,514 | 130 | 0.85 (0.60, 1.19) |
| Four and above births | 4,571 | 513 | 0.85 (0.60, 1.19) |
| **Marital status** | | | |
| Married | 10,191 | 967 | 1 |
| Never married | 273 | 28 | 1.22 (0.78, 1.90) |

(*Continued*)

**Table 5.** (Continued)

| Variable | Abortion | | AOR (95% CI) |
|---|---|---|---|
| | No | Yes | |
| Widowed | 158 | 8 | 0.52 (0.24, 1.11) |
| Divorced | 349 | 24 | 0.78 (0.49, 1.23) |

* = p-value<0.05,

** = p-value<0.01,

AOR: Adjusted Odds Ratio; CI: Confidence Interval.

the risk of poor pregnancy outcome like abortion [43]. Moreover, as maternal age increase, the risk of chromosomal abnormality will be increased and uterine and hormonal function will be decreased, which finally result in miscarriage/abortion if women become pregnant at an older age [44].

Our study revealed that media exposure was a significant predictor associated with increased odds of abortion. This result agrees with reports in Ghana and Mozambique [36]. The possible reason might be due to media is an important mechanism in providing information about how and where to terminate a pregnancy. Furthermore, women who have media exposure might be aware of available laws related to abortion and less likely to be stigmatized by society [45].

The odds of abortion among protestant religious followers were lower compared to Orthodox Christians. It was consistent with a study finding in China [46] and the possible explanation could be due to lack of access to reproductive health services, and deep-rooted cultural belief towards abortion in the community [46]. Regarding wealth status, in this study, women from the richest household had higher odds of experiencing abortion than those from the poorest household. This finding was consistent with studies in Ghana [47] and Nepal [48]. This might be due to the reason that the wealth status of women can determine their ability to cover the cost of maternal health care services. Besides, poor women are facing cost barriers like transportation costs since the abortion services did not perform elsewhere, this can impede women to have an abortion.

In this study, maternal education was a significant predictor of abortion. Women who had attained primary education had higher odds of abortion as compared to women who had no formal education. This was in line with study findings reported in northwest Ethiopia [37], and India [38]. It could be due to the reason that educated women didn't need to have birth to meet the demands of ongoing education [49]. Besides, educated women might have information and access to abortion services [50].

This study has both strengths and limitations. Since the study used nationally representative data, the findings of the study can be generalized at the national level. Besides, the study was based on an advanced (appropriate) model, by taking into account the clustering effect, to get reliable standard error and estimate. However, due to the cross-sectional nature of the data, the temporal relationship can't be established. Besides, since the outcome was sensitive and collected based on self-reporting, there may be a possibility of social desirability bias that can lead to under-reporting.

## Conclusion

This study showed that the spatial distribution of abortion was significantly varied across the country. The hotspot areas of abortion were located in the northern Tigray region, border areas of Oromia, SNNPR, and Amhara region. Besides, maternal age, maternal education,

wealth status, media exposure, religion, and residence were significantly associated with abortion. Therefore, policymakers and governmental and non-governmental organizations could strengthen the effort towards reproductive health services particularly for rural residents and should design effective public health interventions in the identified hotspot areas to reduce the incidence of abortion and abortion-related morbidity and mortality. Besides, we recommend scholars to examine the reason why abortion had significant geographic variation within the countries using a detailed exploration like qualitative study.

## Acknowledgments

We would like to thank the measure DHS program for providing the data set.

## Author Contributions

**Conceptualization:** Getayeneh Antehunegn Tesema, Tesfaye Hambisa Mekonnen, Achamyeleh Birhanu Teshale.

**Data curation:** Getayeneh Antehunegn Tesema, Tesfaye Hambisa Mekonnen, Achamyeleh Birhanu Teshale.

**Formal analysis:** Getayeneh Antehunegn Tesema, Tesfaye Hambisa Mekonnen, Achamyeleh Birhanu Teshale.

**Investigation:** Getayeneh Antehunegn Tesema, Tesfaye Hambisa Mekonnen, Achamyeleh Birhanu Teshale.

**Methodology:** Getayeneh Antehunegn Tesema, Tesfaye Hambisa Mekonnen, Achamyeleh Birhanu Teshale.

**Software:** Getayeneh Antehunegn Tesema, Tesfaye Hambisa Mekonnen, Achamyeleh Birhanu Teshale.

**Validation:** Getayeneh Antehunegn Tesema, Tesfaye Hambisa Mekonnen, Achamyeleh Birhanu Teshale.

**Visualization:** Getayeneh Antehunegn Tesema, Tesfaye Hambisa Mekonnen, Achamyeleh Birhanu Teshale.

**Writing – original draft:** Getayeneh Antehunegn Tesema, Tesfaye Hambisa Mekonnen, Achamyeleh Birhanu Teshale.

**Writing – review & editing:** Getayeneh Antehunegn Tesema, Tesfaye Hambisa Mekonnen, Achamyeleh Birhanu Teshale.

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
