## [Decision Letter · Decision Letter 0]

2 Jan 2020

PONE-D-19-30532

Spatial distribution and determinants of abortion among reproductive-age women in Ethiopia, Evidence from Ethiopian Demographic and Health Survey (EDHS) 2016 data: Spatial and Mixed-effect analysis

PLOS ONE

Dear Mr Tesema,

Thank you for submitting your manuscript to PLOS ONE. After careful consideration, we feel that it has merit but does not fully meet PLOS ONE’s publication criteria as it currently stands. Therefore, we invite you to submit a revised version of the manuscript that addresses the points raised during the review process.

Please be sure to address all issues raised by the reviewers as well as the editor's comments in the attached pdf file.

We would appreciate receiving your revised manuscript by Feb 16 2020 11:59PM. To enhance the reproducibility of your results, we recommend that if applicable you deposit your laboratory protocols in protocols.io, where a protocol can be assigned its own identifier (DOI) such that it can be cited independently in the future. For instructions see: http://journals.plos.org/plosone/s/submission-guidelines#loc-laboratory-protocols

We look forward to receiving your revised manuscript.

Kind regards,

Agricola Odoi, BVM, MSc, PhD, FAHA, FACE

Academic Editor

PLOS ONE

Journal Requirements:

2.  Thank you for stating the following financial disclosure: "N/A"

Please provide an amended Funding Statement that declares *all* the funding or sources of support received during this specific study (whether external or internal to your organization) as detailed online in our guide for authors at http://journals.plos.org/plosone/s/submit-now Please state what role the funders took in the study.  If any authors received a salary from any of your funders, please state which authors and which funder. If the funders had no role, please state: "The funders had no role in study design, data collection and analysis, decision to publish, or preparation of the manuscript."

4.  We note that Figures 1, 3 and 5-7 in your submission contain map images which may be copyrighted.

a.    You may seek permission from the original copyright holder of Figures 1, 3 and 5-7 to publish the content specifically under the CC BY 4.0 license.

Reviewers' comments:

Reviewer's Responses to Questions

**Comments to the Author**

1. Is the manuscript technically sound, and do the data support the conclusions?

Reviewer #1: Yes

Reviewer #2: Partly

Reviewer #3: Yes

Reviewer #4: Yes

2. Has the statistical analysis been performed appropriately and rigorously? 

Reviewer #1: Yes

Reviewer #2: Yes

Reviewer #3: Yes

Reviewer #4: Yes

3. Have the authors made all data underlying the findings in their manuscript fully available?

Reviewer #1: Yes

Reviewer #2: No

Reviewer #3: Yes

Reviewer #4: Yes

4. Is the manuscript presented in an intelligible fashion and written in standard English?

Reviewer #1: Yes

Reviewer #2: Yes

Reviewer #3: No

Reviewer #4: No

5. Review Comments to the Author

Reviewer #1: Generally, this is an interesting manuscript using epidemiological modelling and spatial epidemiology explore the complex subject of abortion in a developing country. This could inform reproductive health services delivery and other unmet needs in family planing in Ethiopia. It would also be of interest to epidemiologist interested in critiquing epidemiological modelling as a couple of methods are explored. The manuscript could be improved by a further read by english proficient person to sort out grammar and spelling issues and flow. Figures need to be redone to publication quality and also consulting a reproductive health expert in Ethiopia to review this manuscript before resubmission would be useful.

In addition, see comments below

In the Abstract,

1. Statistics in the result section could be summarised better, eg not over repeating 95% CI, only present the most important and interesting results eg you don’t have to put all the statistics for all the age groups,

2. Some repetitions in the conclusions

Line 88, what do you mean by wide gap in abortions

Revise some sentences

Line 113 missing full stop

Line 121, do you mean sample size

Statistics, I think one appropriate method of model evaluations and evaluations could be chosen and be used

Line 245-249, two sentences all talking about high prevalence, which regions do you really refer to as high prevalence

Figures are of poor quality, can be improved to publication quality.

Reviewer #2: This manuscript needs several major changes.

First, a thorough grammatical editing is necessary, as the very first sentence has a spelling error ( uretro)

Secondly, the figures are not helping your analysis. Figure 1 is virtually illegible, Figure 2 needs to be a map at the regional level, with correctly capitalized region/state names. Furthermore, Figure 3 needs help, the points are far too small for the reader to understand anything about the spatial variation in the rate, Figure 4 can be eliminated as the moran I statistic can be reported in the text. Adjust the number of decimal places on the maps, there are too many decimal values, you should report at most 2 decimal places.

Regarding your statistical model, it's more common to model the spatial variation using a CAR random effect in the binomial model, and visualize the random effect or the smoothed rate map, versus doing separate kriging or scan statistic methods. Why bother reporting the un-adjusted odd ratios? THis seems pointless to me, as you end up adjusting them anyway. LIkewise, you do not need to report the weighted n in your table 1, just the %'s

Finally, you need to specify a hypothesis, there are no real research questions or testable hypotheses specified.

Reviewer #3: This is an interesting and well written paper documenting clustering of abortion in Ethiopia. The importance of the topic is well framed in the introduction, and the investigators use appropriate methods to draw conclusions.

My only suggestion is that the manuscript receive another read through to make minor edits to sentence structure for clarity.

Reviewer #4: The authors present an interesting examination of the spatial distribution of abortion in Ethiopia using demographic and health survey data. The methods are appropriately used, with one question about whether the residuals were spatially autocorrelated in the model.

1. Lines 25-27, line 63, line 65, line 67, line 69 and throughout. The authors should specify that “unsafe abortion” rather than “abortion” is a major cause of maternal mortality and a public health concern. Right now the two concepts are conflated.

2. Line 33, line 36. Can the authors define abortion in this population? Is this the percentage of women interviewed who ever had an abortion? Or the percentage of previous pregnancies that ended in abortion?

3. Throughout the authors talk about the rate of abortion and the prevalence of abortion. The authors need to define each, which I believe will have different meanings and interpretations. Are these prevalence of women ever having an abortion, prevalence of pregnancies ending in abortion, or what?

4. The authors should limit significant digits on figure 3. 0-5%, 5-15%, 15-28%, 28-50%.

5. The authors did not present the results from the global moran’s I of the residuals of their regression model. Was this non-significant, indicating that the model explained the spatial variance in the outcome? Or did the authors need to adjust their approach to account for spatially correlated data?

6. For figure 5, I presume this is the LISA? Please indicate

7. The authors should limit significant digits on Figure 6 in a same manner as comment 4.

8. For figure seven I suggest using a single color for their hot spots identified (presuming these are all hotspots). They’re all significant.

9. I didn’t see a good subsection in the methods on the variables considered for the regression model. This needs to be better explained. In the table there was no urban/rural, which I would expect to be a significant factor.

10. I think the authors would benefit from a copyeditor for the English.

6. PLOS authors have the option to publish the peer review history of their article (what does this mean?). If published, this will include your full peer review and any attached files.

Reviewer #1: Yes: Luke Nyakarahuka

Reviewer #2: No

Reviewer #3: No

Reviewer #4: No

---

## [Author Response · Author response to Decision Letter 0]

23 Feb 2020

Point by point response for editors and reviewers comments 

Manuscript title: Spatial distribution and determinants of abortion among reproductive-age women in Ethiopia, Evidence from Ethiopian Demographic and Health Survey (EDHS) 2016 data: Spatial and Mixed-effect analysis

Manuscript ID: PONE-D-19-30532

Dear editor/reviewer. 

Dear all,

We would like to thank you for these constructive, building and improvable comments on this manuscript that would improve the substance and content of the manuscript. We considered each comment and clarification questions of editors and reviewers on the manuscript thoroughly. Our point-by-point responses for each comment and questions are described in detail on the following pages. Further, the details of changes were shown by track changes in the supplementary document attached.

Response to editors

Authors’ response: Thank you, editor. We have prepared the manuscript according to PLOS ONE’s style. (see the revised manuscript)

2. Thank you for stating the following financial disclosure: "N/A"

Authors’ response: Thank you, editor. As we have stated in the documents, the study was done based on Ethiopian Demographic and Health Survey (EDHS) which is already available at measure DHS program. We request this program by sending the objectives of the study and we receive an authorization letter from the DHS program.

Authors’ response: We have created ORCID

4. We note that Figures 1, 3 and 5-7 in your submission contain map images which may be copyrighted.

Authors’ response: Thank you, editors, for your concern. The map is not copyrighted rather we have done using GIS software based on the shapefile of Ethiopia received from Ethiopian Central Statistical Agency (CSA) by explaining the purpose of the study and GPS data (longitude and latitude) from measure DHS program by explaining the objective of the study through online requesting and allow us to access the shapefile and GPS data. Now we cite the source of the shapefile since it is needed to explore the spatial distribution of abortion. 

Response to reviewers’ comments 

 Reviewer # 1

1. Statistics in the result section could be summarised better, eg not over repeating 95% CI, only present the most important and interesting results eg you don’t have to put all the statistics for all the age groups,

Authors’ response: Thank you, reviewer, for your valuable comment. We accepted and corrected it. (See the revised manuscript) 

2. Some repetitions in the conclusions

Authors’ response: Thank you, reviewer. We rewrite the conclusion section and remove the repetitions. (See the revised manuscript)

• Line 88, what do you mean by the wide gap in abortions

Authors’ response: Thank you, reviewer. We have stated in the Introduction section as there is wide gap in abortions across and within countries meanwhile we reviewed kinds of literature conducted on abortion the prevalence of abortion varied across countries and within countries and this gives an insight that abortion has been varied across countries and spatial study is needed to identify which areas are significant hotspot. 

• Revise some sentences, Line 113 missing full stop, Line 121, do you mean sample size

Authors’ response: Thank you, reviewer. We have corrected it. (See the revised manuscript) 

• Statistics, I think one appropriate method of model evaluations and evaluations could be chosen and be used

Authors’ response: Thank you, reviewer. We used AIC, BIC, and Deviance for model comparison but mainly we depend on Deviance since our models are nested models and we used ICC, AIC, and BIC as supportive.

• Line 245-249, two sentences all talking about high prevalence, which regions do you refer to them as high prevalence

Authors’ response: Thank you, reviewer, for your comments. It was an editorial error and the second one was areas with a low prevalence of abortion, we had modified in the revised document. (see the revised document)

• Figures are of poor quality, can be improved to publication quality.

Authors’ response: Thank you, reviewer, we had improved the figure quality using PACE in TIFF format (see the revised document)

Reviewer#2 #2: 

1. First, a thorough grammatical editing is necessary, as the very first sentence has a spelling error (uretro)

Authors’ response: Thank you, reviewer, for the valuable comments. We accepted the comments and we corrected the grammatical errors extensively the entire manuscript. For Uretro it is corrected as the uterus (see the revised manuscript)

2. Secondly, the figures are not helping your analysis. Figure 1 is virtually illegible, Figure 2 needs to be a map at the regional level, with correctly capitalized region/state names. Furthermore, Figure 3 needs help, the points are far too small for the reader to understand anything about the spatial variation in the rate, Figure 4 can be eliminated as the Moran I statistic can be reported in the text. Adjust the number of decimal places on the maps, there are too many decimal values, you should report at most 2 decimal places.

Authors’ response: Thank you, reviewer, for your valuable comments. We were included Figure 1 to show the study area on the map and as you said it is not that much important and we removed in the revised manuscript. We have modified Figure 2 in ascending order based on the prevalence of abortion across regions by capitalizing on the name of the region. For Figure 3 the points were placed in decimal places and know we put in terms of percentage of abortion at enumeration areas by reducing the decimal places. We had removed Figure 4 as it is well stated in the form of text. (See the revised manuscript)

• Regarding your statistical model, it's more common to model the spatial variation using a CAR random effect in the binomial model, and visualize the random effect or the smoothed rate map, versus doing separate kriging or scan statistic methods. 

Authors’ response: Thank you, reviewer, for the comments. We have done Kriging interpolation analysis for predicting the prevalence of abortion in unsampled areas based on observed data and SaTScan analysis to identify hotspot areas of abortion by running circular windows but we haven't done the Conditional Autoregressive (CAR) model and visualize the random effect because there are no covariates collected at Enumeration Area (EAs) level in EDHS data. Since the GPS data were collected at EA level but the covariates were collected at the individual level in EDHS data that is why we didn't do the CAR model to visualize the random effects with covariates.

• Why bother reporting the un-adjusted odd ratios? THis seems pointless to me, as you end up adjusting them anyway. Like wise, you do not need to report the weighted n in your table 1, just the %'s

Authors’ response: Thank you, reviewer. We accepted the comments and we removed the COR and the weighted n in table 1. (See the revised manuscript)

• Finally, you need to specify a hypothesis, there are no real research questions or testable hypotheses specified. 

Authors’ response: Thank you, reviewer. The research questions in this study were 

1. Whether the spatial distribution of abortion is random or not? Answered by Global spatial autocorrelation test (Moran’s Index) 

2. Where are the significant hotspot areas of abortion in Ethiopia? Answered by SaTScan analysis

3. What are the factors that are significantly associated with abortion? Answered by GLMM (mixed-effect logistic regression analysis) 

Reviewer #3

 This is an interesting and well written paper documenting clustering of abortion in Ethiopia. The importance of the topic is well framed in the introduction, and the investigators use appropriate methods to draw conclusions.

Authors’ response: Thank you, reviewer.

My only suggestion is that the manuscript receive another read through to make minor edits to sentence structure for clarity.

Authors’ response: Thank you, reviewer. We extensively edit sentence structures and grammar with the help of language experts. (See the revised manuscript) 

Reviewer #4

The authors present an interesting examination of the spatial distribution of abortion in Ethiopia using demographic and health survey data. The methods are appropriately used, with one question about whether the residuals were spatially autocorrelated in the model.

Authors’ response: Thank you, reviewer. We analyzed global spatial autocorrelation using Moran's Index and was significant. It revealed that the spatial distribution of abortion was non-random with Global Moran's I 0.06 (p<0.001) (significant spatial dependence). (See the revised manuscript)

1. Lines 25-27, line 63, line 65, line 67, line 69 and throughout. The authors should specify that “unsafe abortion” rather than “abortion” is a major cause of maternal mortality and a public health concern. Right now the two concepts are conflated.

Authors’ response: Thank you, reviewer. We have corrected as unsafe abortion in the document but in EDHS the data was collected as abortion it was not separately recorded as unsafe and safe abortion. (See the revised manuscript)

2. Line 33, line 36. Can the authors define abortion in this population? Is this the percentage of women interviewed who ever had an abortion? Or the percentage of previous pregnancies that ended in abortion?

Authors’ response: Thank you, reviewer. For this study, we define "abortion as the percentage of previous pregnancy that ended in abortion". (See the revised manuscript)

3. Throughout the authors talk about the rate of abortion and the prevalence of abortion. The authors need to define each, which I believe will have different meanings and interpretations. Are these prevalence of women ever having an abortion, prevalence of pregnancies ending in abortion, or what?

Authors’ response: Thank you, reviewer. You are right the rate of abortion and prevalence of abortion is different in meaning and interpretations. When we say the rate of abortion it is defined as the number of pregnancy ended in abortion per 1000 pregnancy whereas the prevalence of abortion is defined as the percentage of abortion per 100 pregnancy. For this study, we had reported the prevalence of abortion. (See the revised manuscript)

4. The authors should limit significant digits on figure 3. 0-5%, 5-15%, 15-28%, 28-50%.

Authors’ response: Thank you, reviewer. We have modified the maps as you recommend us. (See the revised manuscript)

5. The authors did not present the results from the global moran’s I of the residuals of their regression model. Was this non-significant, indicating that the model explained the spatial variance in the outcome? Or did the authors need to adjust their approach to account for spatially correlated data?

Authors’ response: Thank you, reviewer. We didn't do the spatially weighted regression since there is no covariate collected at the EA level. We did only the Global spatial autocorrelation, spatial interpolation, and SaTScan analysis. 

6. For figure 5, I presume this is the LISA? Please indicate

Authors’ response: Thank you, reviewer. Figure 5 was Local indicators of spatial autocorrelation (LISA) using Getis Ord Gi statistics of hotspot analysis to identify significant hotspot areas and significant cold spot areas of abortion. As per the editors' comment, we had removed Figure 5 since we used SaTScan analysis to identify significant hotspot areas of abortion and this is very informative from a public health perspective.

7. The authors should limit significant digits on Figure 6 in a same manner as comment 4.

Authors’ Response: Thank you, reviewer. We have corrected by limiting significant digits. (See the revised Figure)

8. For figure seven I suggest using a single color for their hot spots identified (presuming these are all hotspots). They’re all significant.

Authors’ response: Thank you, reviewer. We have modified the figure as you suggest and we have clipped within the study area. (See the revised Figure)

9. I didn’t see a good subsection in the methods on the variables considered for the regression model. This needs to be better explained. In the table there was no urban/rural, which I would expect to be a significant factor.

Authors’ response: Thank you, reviewer. We have stated the variables considered for the regression model in the revised manuscript. The place of residence was one of the significant factors associated with abortion and it is already found in the regression table. (See the revised manuscript)

10. I think the authors would benefit from a copyeditor for the English.

Authors’ response: Thank you, reviewer. We had extensively edit the grammar and sentence structure with the help of Language experts. (See the revised manuscript)

---

## [Decision Letter · Decision Letter 1]

18 May 2020

PONE-D-19-30532R1

Spatial distribution and determinants of abortion among reproductive-age women in Ethiopia, Evidence from Ethiopian Demographic and Health Survey (EDHS) 2016 data: Spatial and Mixed-effect analysis

PLOS ONE

Dear Mr Tesema,

Thank you for submitting your manuscript to PLOS ONE. After careful consideration, we feel that it has merit but does not fully meet PLOS ONE’s publication criteria as it currently stands. Therefore, we invite you to submit a revised version of the manuscript that addresses the points raised during the review process.

Specifically, please address the issues raised in the attached pdf file. Additionally, the manuscript has numerous grammatical errors that render it not suitable for publication in its current form. Therefore, I strongly recommend that you get it reviewed and thoroughly edited by a native English speaker.

We would appreciate receiving your revised manuscript by Jul 02 2020 11:59PM. To enhance the reproducibility of your results, we recommend that if applicable you deposit your laboratory protocols in protocols.io, where a protocol can be assigned its own identifier (DOI) such that it can be cited independently in the future. For instructions see: http://journals.plos.org/plosone/s/submission-guidelines#loc-laboratory-protocols

We look forward to receiving your revised manuscript.

Kind regards,

Agricola Odoi, BVM, MSc, PhD, FAHA, FACE

Academic Editor

PLOS ONE

Reviewers' comments:

Reviewer's Responses to Questions

**Comments to the Author**

1. If the authors have adequately addressed your comments raised in a previous round of review and you feel that this manuscript is now acceptable for publication, you may indicate that here to bypass the “Comments to the Author” section, enter your conflict of interest statement in the “Confidential to Editor” section, and submit your "Accept" recommendation.

Reviewer #1: All comments have been addressed

Reviewer #3: All comments have been addressed

Reviewer #4: (No Response)

2. Is the manuscript technically sound, and do the data support the conclusions?

Reviewer #1: Yes

Reviewer #3: Yes

Reviewer #4: Yes

3. Has the statistical analysis been performed appropriately and rigorously? 

Reviewer #1: Yes

Reviewer #3: Yes

Reviewer #4: Yes

4. Have the authors made all data underlying the findings in their manuscript fully available?

Reviewer #1: Yes

Reviewer #3: Yes

Reviewer #4: Yes

5. Is the manuscript presented in an intelligible fashion and written in standard English?

Reviewer #1: Yes

Reviewer #3: Yes

Reviewer #4: Yes

6. Review Comments to the Author

Reviewer #1: (No Response)

Reviewer #3: I have no further comments on this manuscript. All of my comments have been addressed in this revised manuscript.

Reviewer #4: (No Response)

7. PLOS authors have the option to publish the peer review history of their article (what does this mean?). If published, this will include your full peer review and any attached files.

Reviewer #1: Yes: Luke Nyakarahuka

Reviewer #3: No

Reviewer #4: No

---

## [Author Response · Author response to Decision Letter 1]

4 Jun 2020

PLOS ONE

Point by point response for editors/reviewers comments 

Manuscript title: Spatial distribution and determinants of abortion among reproductive-age women in Ethiopia, Evidence from Ethiopian Demographic and Health Survey (EDHS) 2016 data: Spatial and Mixed-effect analysis

Manuscript ID: PONE-D-19-30532R1

Dear editor/reviewer. 

Dear all,

We would like to thank you for this constructive, building, and improvable comments on this manuscript that would improve the substance and content of the manuscript. Further, the details of changes we made is shown by track changes in the supplementary document attached. The manuscript language was checked by language professionals and we follow journal guideline. Response to Editors comments 

1. Specifically, please address the issues raised in the attached pdf file. Additionally, the manuscript has numerous grammatical errors that render it not suitable for publication in its current form. Therefore, I strongly recommend that you get it reviewed and thoroughly edited by a native English speaker. 

Authors’ response: Thank you Editor for the comments. We had extensively revised the manuscript. The grammar and editorial errors was extensively edited, and reviewed thoroughly with the help of language experts working at university of Gondar (See the revised manuscript).

---

## [Editor Report · Decision Letter 2]

16 Jun 2020

Spatial distribution and determinants of abortion among reproductive age women in Ethiopia, Evidence from Ethiopian Demographic and Health Survey 2016 data: Spatial and Mixed-effect analysis

PONE-D-19-30532R2

Dear Dr. Tesema,

We’re pleased to inform you that your manuscript has been judged scientifically suitable for publication and will be formally accepted for publication once it meets all outstanding technical requirements.

Kind regards,

Agricola Odoi, BVM, MSc, PhD, FAHA, FACE

Academic Editor

PLOS ONE
---

## [Editor Report · Acceptance letter]

19 Jun 2020

PONE-D-19-30532R2 

Spatial distribution and determinants of abortion among reproductive age women in Ethiopia, Evidence from Ethiopian Demographic and Health Survey 2016 data: Spatial and Mixed-effect analysis 

Dear Dr. Tesema:

I'm pleased to inform you that your manuscript has been deemed suitable for publication in PLOS ONE. Congratulations! Your manuscript is now with our production department. 

Kind regards, 

on behalf of

Prof. Agricola Odoi 

Academic Editor

PLOS ONE